# Incorporation of Incompatible Strontium and Barium Ions into Calcite (CaCO$_3$) through Amorphous Calcium Carbonate

**Ayaka Saito [1], Hiroyuki Kagi [1,\*], Shiho Marugata [1], Kazuki Komatsu [1], Daisuke Enomoto [1], Koji Maruyama [1] and Jun Kawano [2]**

[1] Geochemical Research Center, Graduate School of Science, The University of Tokyo, 7-3-1 Hongo, Tokyo 113-0033, Japan; ayawaka921105@gmail.com (A.S.); marugata@eqchem.s.u-tokyo.ac.jp (S.M.); kom@eqchem.s.u-tokyo.ac.jp (K.K.); enomoto.daisuke@iri-tokyo.jp (D.E.); k6.maru@gmail.com (K.M.)

[2] Department of Earth and Planetary Sciences, Faculty of Science, Hokkaido University, N10 W8, Kita-ku, Sapporo 060-0810, Japan; j-kawano@sci.hokudai.ac.jp

\* Correspondence: kagi@eqchem.s.u-tokyo.ac.jp

**Abstract:** Calcite is a ubiquitous mineral in nature. Heavy alkaline-earth elements with large ionic radii such as Sr$^{2+}$ and Ba$^{2+}$ are highly incompatible to calcite. Our previous study clarified that incompatible Sr$^{2+}$ ions can be structurally incorporated into calcite through crystallization from amorphous calcium carbonate (ACC). In this study, we synthesized Sr-doped calcite with Sr/(Sr + Ca) up to 30.7 ± 0.6 mol% and Ba-doped calcite with Ba/(Ba + Ca) up to 68.6 ± 1.8 mol%. The obtained Ba-doped calcite samples with Ba concentration higher than Ca can be interpreted as Ca-containing barium carbonates with the calcite structure which have not existed so far because barium carbonate takes the aragonite structure. X-ray diffraction (XRD) patterns of the Sr-doped and Ba-doped calcite samples obtained at room temperature showed that reflection 113 gradually weakened with increasing Sr/(Sr + Ca) or Ba/(Ba + Ca) ratios. The reflection 113 disappeared at Ba/(Ba + Ca) higher than 26.8 ± 1.6 mol%. Extinction of reflection 113 was reported for pure calcite at temperatures higher than 1240 K, which was attributed to the rotational (dynamic) disorder of CO$_3^{2-}$ in calcite. Our Molecular Dynamics (MD) simulation on Ba-doped calcite clarified that the CO$_3^{2-}$ ions in Ba-doped calcites are in the static disorder at room temperature. The CO$_3^{2-}$ ions are notable tilted and displaced from the equilibrium position of pure calcite.

**Keywords:** calcite; amorphous calcium carbonate; disorder; MD simulation

## 1. Introduction

Calcium carbonate (CaCO$_3$) is a ubiquitous mineral in nature that has three anhydrous polymorphs: calcite, aragonite, and vaterite. Calcite has a trigonal structure and it is thermodynamically stable under ambient condition. Aragonite, an orthorhombic structure, is a high-pressure phase. Vaterite is a metastable phase and it is a rare mineral compared to calcite and aragonite. In addition to these three anhydrous polymorphs, amorphous calcium carbonate (hereafter amorphous calcium carbonate (ACC); CaCO$_3 \cdot n$H$_2$O, $n < 1.5$) is known to exist [1]. In living organisms, ACC is mainly used as a precursor for the formation of crystalline calcium carbonate and either has an aragonite-like short-range order or a calcite-like short-range order. Apart from biominerals, synthetically prepared ACCs have calcite-like short range structures [2,3]. Since ACC is thermodynamically metastable, it is easily crystallized at high temperature or high pressure [4,5].

Calcium carbonate exists mostly as calcite and aragonite in the natural environment. It is known that the structures of carbonates depend on the ionic radii of divalent metal ions [6]. Divalent ions with an ionic radius smaller than that of $Ca^{2+}$ (1.00 Å) form a carbonate with the calcite structure. On the other hand, divalent ions with an ionic radius larger than that of $Ca^{2+}$ form a carbonate with the aragonite structure. The incorporation of impurity ions into structures of carbonates also depends on the radii of impurity ions. Carbonates with the calcite structure tend to capture impurity elements of which the ionic radius is smaller than that of calcium ion. In contrast, carbonates with the aragonite structure tend to capture impurity elements of which the ionic radius is larger than that of $Ca^{2+}$. From the above, divalent metal ions such as $Sr^{2+}$ and $Ba^{2+}$ are incompatible to calcite. Pingitore and Eastman [7] studied the behavior of trace elements, $Sr^{2+}$ and $Ba^{2+}$, into calcite. Partition coefficients of $Ba^{2+}$ and $Sr^{2+}$ were $k_{calcite}^{Ba^{2+}} = 0.6 \pm 0.01$ and $k_{calcite}^{Sr^{2+}} = 0.05 \sim 0.14$, respectively. Therefore, $k_{calcite}^{Ba^{2+}} < k_{calcite}^{Sr^{2+}}$ is consistent with the size of ionic radii showing that Ba is more incompatible to calcite than Sr.

Matsunuma et al. reported that high concentration of incompatible Sr was captured into the structure of calcite by pressurizing ACC [8]. The unit-cell volume of calcite samples precipitated from a supersaturated solution showed no noteworthy increase. During crystal growth from the supersaturated solution, Sr is excluded from calcite because of its incompatibility. In contrast, the unit-cell volume of calcite samples obtained from crystallization from ACC notably increased with increasing Sr concentration in starting solutions [8]. Because ACC has significant flexibility in the structure, impurity ions with high incompatibility to calcite can be captured in ACC. Rapid crystallization from ACC may result in the introduction of highly incompatible ions into calcite. Not only being limited to calcite, impurity doping via amorphous state to a crystallite phase can be applied to other substances.

In this study, we verified the Sr/(Sr + Ca) in calcite prepared from Sr-doped ACC samples. Also, incorporation of more incompatible $Ba^{2+}$ (1.35 Å) into calcite was investigated. Changes in properties of Sr-doped and Ba-doped calcite were investigated as well.

## 2. Experimental Procedures

### 2.1. Synthesis of Sr-Doped Calcite and Ba-Doped Calcite

Strontium-doped ACC samples were synthesized based on Matsunuma et al. [8]. Ice-cooled 0.1 M $Na_2CO_3$ and blended 0.1 M solutions of $CaCl_2$ and $SrCl_2$ with varying Sr/(Sr + Ca) from 0 to 50 mol% were mixed at 0 °C with the weight ratio of 1:1. The initial pH value of the mixed solution is approximately 11.5. White precipitates were obtained after mixing the solutions in the reaction vessel. The precipitates were immediately filtered using a membrane filter (0.45 μm) with vacuum filtration. Obtained precipitates were then washed with acetone and dried at 25 °C for 1 day in a vacuum desiccator evacuated with a diaphragm pump. The reasons why the sample solutions were kept at 0 °C and the obtained ACC samples were rinsed with acetone are to avoid crystallization of calcite. Synthesized Sr-doped ACC samples were pressurized using a hydraulic press in a tungsten carbide (WC) piston-cylinder of 4 mm inner diameter at 0.8 GPa for 10 min at room temperature for pressure-induced crystallization of calcite. After decompression, the samples were recovered and kept in a vacuum desiccator to remove water that emitted after pressure-induced crystallization. In addition, calcite samples were also prepared by heating Sr-doped ACC samples at 400 °C for 2 h.

Barium-doped ACC samples were also prepared by the same method as Sr-doped ACC samples, mixing ice-cooled 0.1 M $Na_2CO_3$ and blended solutions of $CaCl_2$ and $BaCl_2$ with varying Ba/(Ba + Ca) from 0 to 80 mol% with the weight ratio of 1:1. After the samples were dried for 1 day in a vacuum desiccator, Ba-doped ACC samples were heated for 2 h at 400 °C for heat-induced crystallization of calcite.

*2.2. Sample Analysis*

Powder X-ray diffraction (XRD) patterns of the obtained samples were measured using a silicon zero background plate and an X-ray diffractometer (Miniflex II, Rigaku Corp., Tokyo, Japan). The measurement conditions for XRD were 0.02° step, scanned region from 10° to 70° in 2$\theta$ with a scan rate of 1° per minute, Cu$K\alpha$ radiation operated at 15 mA and 30 kV. For refinement of lattice parameters of calcite samples, potassium chloride powder was mixed with the samples as an internal standard for lattice constant and Rietveld analysis was conducted using general structure analysis system (GSAS) software and EXPGUI [9,10]. Field emission Scanning electron microscope/Energy dispersive X-ray spectroscopy (SEM-EDS) analysis (JSM-7000, JEOL, Tokyo, Japan) was conducted to observe the morphology of impurity-doped calcite samples and to determine the Sr and Ba concentration in the synthesized samples. The pelletized calcite samples were fixed on a glass slide with carbon tapes and carbon coating was applied on the samples.

## 3. Molecular Dynamics (MD) Simulations of Ba-Doped Calcite

MD simulations of Ba-doped calcite were conducted to investigate the behavior of carbonate ions in Ba-doped calcite at an atomistic level. Because the ionic radius of Ba is larger than that of Sr, Ba-doped calcite should be more suitable to analyze the effect of captured divalent cation on the crystal structure with MD simulation. The interatomic potential function between two atoms (the *i* th and *j* th atoms) used in the present MD calculations is described as follows:

$$\phi_{ij}\left(r_{ij}\right) = \frac{z_i z_j e^2}{r_{ij}} + f_0\left(B_i + B_j\right)exp\frac{A_i + A_j - r_{ij}}{B_i + B_j} - \frac{C_i C_j}{r_{ij}^6} + D_{1ij}exp\left(-\beta_{1ij}r_{ij}\right) + D_{2ij}exp\left(-\beta_{2ij}r_{ij}\right).$$

The equation consists of the Coulombic interaction between point charges, short range repulsion, van der Waals attraction, and Morse potential terms. Here $r_{ij}$ is the interatomic distance between the *i* th and *j* th atoms, $f_0 = 6.9511 \times 10^{-11}$ N is a constant, e is the electronic charge. *z*, *A*, *B*, and *C* are the parameters for each atomic species, and $D_1$, $D_2$, $\beta_1$, and $\beta_2$ are the parameters for the C–O pair.

MD simulations in this study were carried out with the MD program MXDTRICL [11]. The equations of motions were integrated by Verlet's algorithm with a time step of 0.5 fs. Periodic boundary conditions were applied on the MD basic cell. The temperatures and pressures were controlled by scaling particle velocities and simulating the cell parameters, respectively.

The potential parameters set for Ba-doped calcite was derived based on Kawano et al. (see Table 1): the same parameters for O, C, and Ca were used as these determined in Kawano et al. [12] for the phase transition in calcite at high temperatures, and the parameters for Ba was newly derived empirically to reproduce the crystal structure, thermal expansivity, and compressibility of witherite (BaCO$_3$ with aragonite structure) (see Supplementary Materials).

**Table 1.** Parameter sets for interatomic potential function for Molecular Dynamics (MD) simulations. Parameters for Ca, C, O, and O–C are from Kawano et al. [12]. Parameters for Ba was determined in this study.

| Atoms | Z (e) | A (Å) | B (Å) | C (kcal$^{1/2}$ Å3 mol$^{-1/2}$) |
|---|---|---|---|---|
| O | −0.915 | 1.8836 | 0.1658 | 23.351 |
| C | 1.045 | 0.4638 | 0.0784 | 0 |
| Ca | 1.7 | 1.4466 | 0.1042 | 10.086 |
| Ba | 1.7 | 1.65 | 0.102 | 13 |

| Atomic Pair | D1 (kJ mol$^{-1}$) | β1 (Å$^{-1}$) | D2 (kJ mol$^{-1}$) | β2 (Å$^{-1}$) |
|---|---|---|---|---|
| O–C | 45735 | 5.14 | −4936.066 | 2.57 |

The unit cell adopted in the MD simulation of Ba-doped calcite was composed of 72 crystallographic unit cells of calcite ($a_{MD} = 6a$, $c_{MD} = 2c$, in the hexagonal setting), containing 2160 atoms. Barium ions were randomly substituted to Ca ions in the calcite structure relaxed at 300 K and 1 atm to form initial structure of Ba-doped calcite with various Ba concentration. Atomic behavior was analyzed based on the results of calculations for 20–30 ps (= 40,000–60,000 steps), after preliminary annealing of the initial structure for at least 20 ps (= 40,000 steps) which is ensured to be long enough to equilibrate the system. Simulations at high temperature were started from the relaxed structure of Ba-doped calcite with the targeted Ba concentration, which were annealed at 300 K and 1 atom. The diffracted intensity was calculated from the structure factor that was directly obtained from the MD simulated atomic positions by using a program developed by Miyake et al. [13].

## 4. Results and Discussion

### 4.1. Increase of Lattice Parameters of Calcite Induced by Incorporation of Sr and Ba

Figure 1a displays XRD patterns of calcite samples obtained from Sr-doped ACC samples after pressure treatment at 0.8 GPa. Calcite was observed as the crystalline phase after pressure treatment on the ACC samples precipitated from solutions with Sr/(Sr + Ca) < 35 mol%. However, strontianite (SrCO$_3$ with the aragonite structure) was observed as a crystalline phase in addition to calcite after pressure treatment on ACC obtained from starting solutions with Sr/(Sr + Ca) > 40 mol%. In contrast, after heating treatment at 400 °C, no strontianite was observed from calcite samples crystallized from ACC obtained from starting solutions with Sr/(Sr + Ca) = 40 mol%. Figure 1b shows XRD patterns of calcite samples obtained by heating treatment on Sr-doped ACC samples. ACC samples precipitated from solutions with Sr/(Sr + Ca) < 40 mol% crystallized into calcite. In contrast, ACC samples from solutions with Sr/(Sr + Ca) = 40 mol% crystallized into strontianite and calcite in a similar way to the case of pressure-induced crystallization.

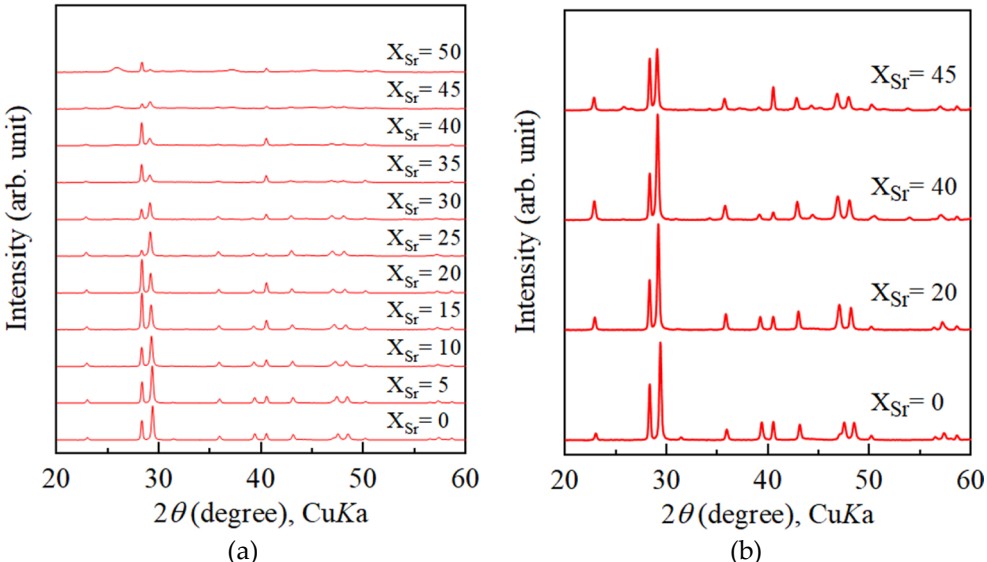

**Figure 1.** Powder X-ray diffraction patterns of calcium carbonate samples precipitated from solutions with various Sr concentrations. (**a**) Crystallization was induced by pressure treatment. (**b**) Powder X-ray diffraction patterns of calcium carbonate samples precipitated from solutions with various Sr concentrations. Crystallization was induced by heating treatment at 400 °C for 2 h. x$_{Sr}$ stands for Sr/(Sr + Ca) of starting solutions.

The unit-cell parameters of calcite samples obtained from heating treatment and pressure treatment were obtained from Rietveld refinement on the XRD patterns. The obtained unit-cell volume data are presented in Figure 2.

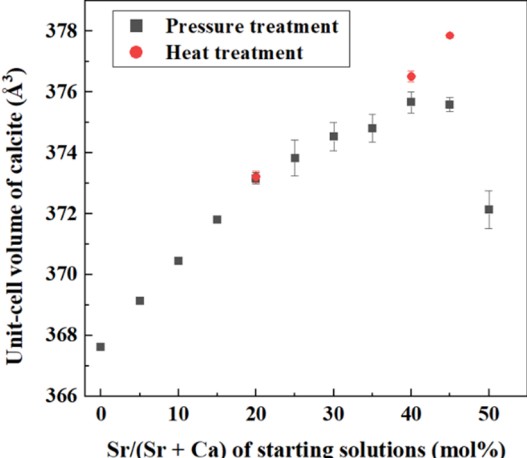

**Figure 2.** Unit-cell volume of calcite samples obtained from heating treatment and pressure treatment vs. Sr/(Sr + Ca) of starting solutions. The sample with the largest volume higher than 377 Å$^3$ contained strotntianite. The error bar was derived from standard deviation of unit cell volume estimated using general structure analysis system (GSAS).

Matsunuma et al. reported the unit-cell volume of calcite obtained by pressurization of ACC increased monotonically up to 372.5 Å$^3$ [8]. As shown in Figure 2, the unit-cell volume of calcite obtained by heating treatment on Sr-doped ACC increased up to 376.51 ± 0.18 Å$^3$, which extended to higher value than the previous study [8]. The unit-cell volume of calcite obtained from pressurization of ACC increased with increasing Sr concentration of starting solutions, but the unit-cell volume drops after the appearance of strontianite (see Figure 2). This indicates that Sr doped in the initial ACC was partitioned to strontianite and Sr concentration in calcite notably decreased.

Figure 3 displays powder XRD patterns of calcium carbonate samples obtained from heating treatment on Ba-doped ACC samples. ACC samples in the range of Ba/(Ba + Ca) = 0 to Ba/(Ba + Ca) = 70 mol% crystallized into exclusively calcite. Samples crystallized from solutions with Ba/(Ba + Ca) > 75 mol% contained witherite (BaCO$_3$ with the aragonite structure) in addition to calcite. Unit-cell parameters of calcite samples obtained from heating treatment on Ba-doped ACC were refined from the XRD patterns. The *c/a* axial ratio increased as the Ba concentration of starting solutions increased (data not shown). This trend also support that Ba is contained in the crystal structure of calcite [6,14]. Figure 4 shows that unit-cell volume of calcite crystallized from Ba-doped ACC monotonically increased with increasing Ba concentration in the starting solutions and the maximum unit-cell volume of Ba-doped calcite was 426.26 ± 0.08 Å$^3$. This result is much higher than that of Sr-doped calcite (376.51 ± 0.18 Å$^3$).

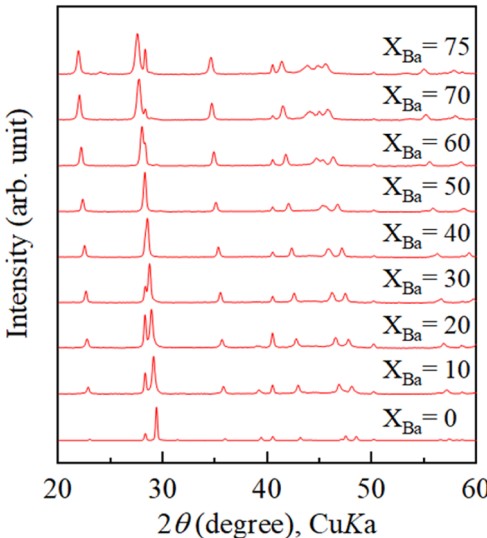

**Figure 3.** Powder X-ray diffraction patterns of calcium carbonate samples with various Ba concentrations obtained from heating treatment. Crystallization was induced by heating treatment at 400 °C for 2 h. $x_{Ba}$ stands for Ba/(Ba + Ca) of starting solutions.

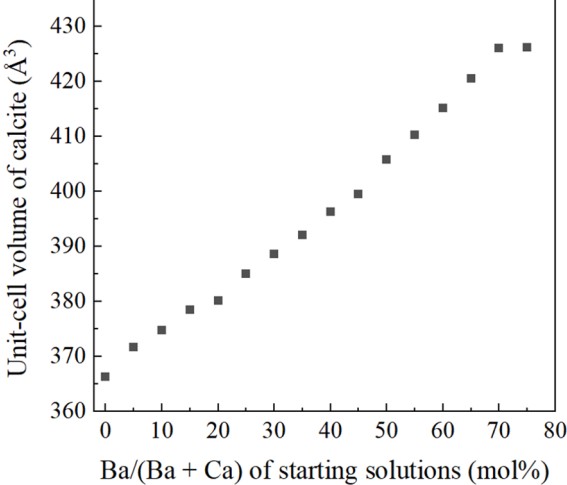

**Figure 4.** Unit-cell volume of calcite samples obtained from heating treatment vs Ba/(Ba + Ca) ratio of starting solutions.

### 4.2. Growth Texture of Sr-Doped Calcite and Ba-Doped Calcite

Figure 5a,b show SEM images of Sr-doped calcite obtained from pressure treatment and heating treatment, and Figure 5c is an SEM image of Ba-doped calcite obtained from the heating treatment. Calcite crystals obtained from pressure treatment have euhedral shapes and the grain size is in the order of 10 µm. In contrast, the grain size of calcite samples obtained from heating treatment was in the order of several tens of nanometers which is much smaller than those obtained from pressure treatment. In pressure-induced crystallization, coexisting fluid plays an important role in crystallization of calcite carbonate; dissolution-recrystallization from coexisting fluid can result in the crystallization of calcite with larger grain size [8,15]. On the other hand, the fine calcite crystals were obtained by heating treatment. During the heating process, the nucleation density and the nucleation rate were high. The contrastive growth textures imply the notable difference in the nucleation density and growth kinetics of calcite between the heating treatment and pressure-treatment.

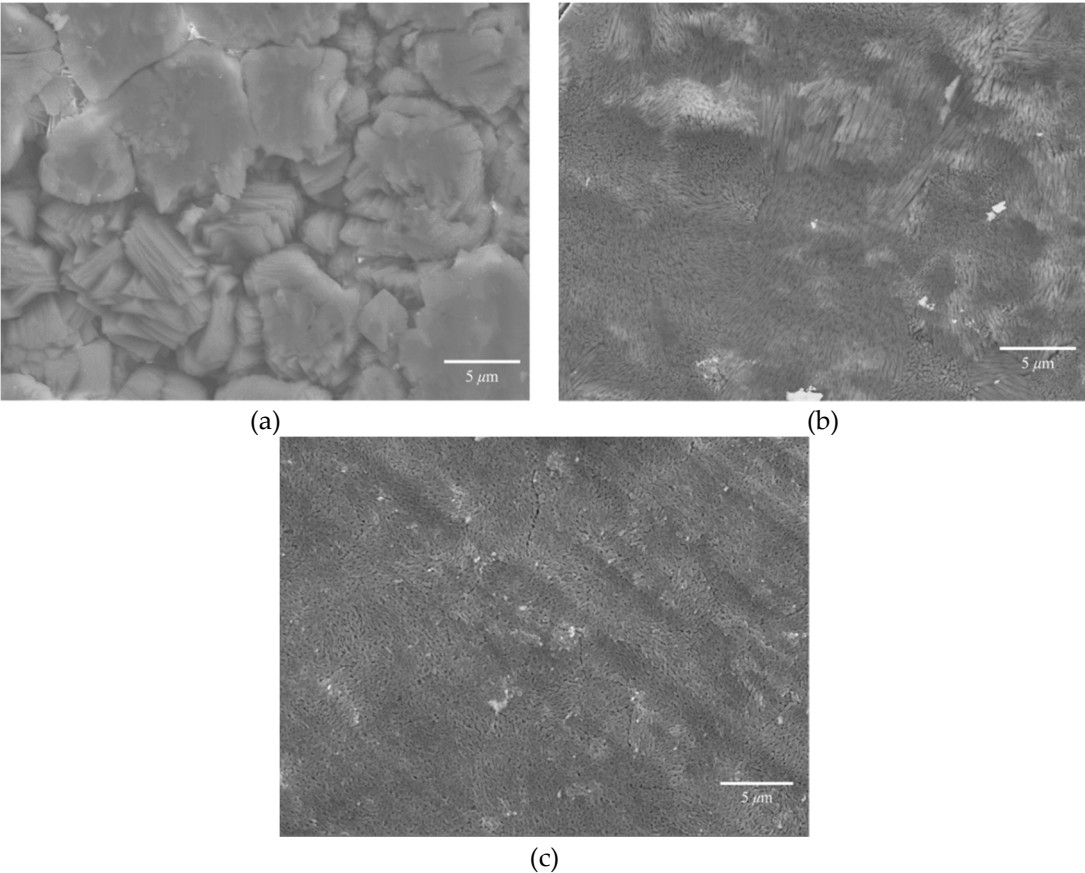

**Figure 5.** (**a**) SEM image of pressurized Sr-doped ACC sample (Sr/(Sr + Ca) = 0.05). (**b**) SEM image of heated Sr-doped ACC sample (Sr/(Sr + Ca) = 0.2). (**c**) SEM image of heated Ba-doped ACC sample (Ba/(Ba + Ca) = 0.45).

*4.3. Incorporation of Sr and Ba into Calcite Lattice*

Strontium concentration of calcite samples was determined from Energy-dispersive X-ray spectroscopy (EDS) measurements. The maximum Sr concentration of calcite obtained by heating Sr-doped ACC was 30.7 ± 0.6 mol% (see Table 2), which was larger than the pressurized sample (24.8 ± 0.4 mol%). These crystallized samples contained no strontianite as an impurity. Strontium concentration of natural samples such as corals and foraminiferas are less than 1 mol% [16,17] and the synthesized samples that were directly precipitated contain about 3 mol% of Sr at most [8]. It is obvious that the Sr concentration obtained in this study is remarkably higher.

**Table 2.** Sr/(Sr + Ca) of calcite samples determined from Energy-dispersive X-ray spectroscopy (EDS) measurements. The calcite samples listed in this table contained no strontianite ($SrCO_3$).

| Sr/(Sr + Ca) in Starting Solutions (mol%) | Sr/(Sr + Ca) in Calcite Determined from EDS Measurements (mol%) | |
|:---:|:---:|:---:|
| | **Pressure Treatment** | **Heat Treatment** |
| 5 | 5.1 ± 0.1 | |
| 10 | 9.5 ± 0.2 | |
| 15 | 14.2 ± 0.2 | |
| 20 | 19.4 ± 0.2 | 19.3 ± 0.7 |
| 40 | | 30.7 ± 0.6 |

In the case of Ba incorporation, the maximum Ba concentration of heat-induced Ba-doped calcite was 68.6 ± 1.8 mol% (see Table 3). This result is much higher than the maximum Sr concentration in calcite. Barium carbonate ($BaCO_3$; witherite) takes an aragonite structure because the ionic radius of $Ba^{2+}$ (1.35 Å) is much larger than $Ca^{2+}$ (1.00 Å) and $Sr^{2+}$ (1.18 Å). Ba-doped calcite obtained in this study has a calcite structure though they occupy more than half $Ba^{2+}$ than $Ca^{2+}$ in the crystal structure, which would be a Ca-doped $BaCO_3$ with a calcite structure. This study clarified that crystallization through ACC made it possible to synthesize such new material.

**Table 3.** Ba/(Ba + Ca) of calcite samples determined from EDS measurements.

| Ba/(Ba + Ca) in Starting Solutions (mol%) | Ba/(Ba + Ca) in Calcite Determined from EDS Measurements (mol%) |
|:---:|:---:|
| 5 | 8.2 ± 0.8 |
| 10 | 13.1 ± 1.1 |
| 15 | 17.6 ± 0.6 |
| 20 | 22.2 ± 1.0 |
| 25 | 26.8 ± 1.6 |
| 30 | 31.7 ± 1.4 |
| 35 | 36.5 ± 1.2 |
| 40 | 39.6 ± 1.7 |
| 45 | 43.0 ± 1.6 |
| 50 | 50.2 ± 1.6 |
| 55 | 53.2 ± 1.2 |
| 60 | 57.6 ± 1.3 |
| 65 | 63.3 ± 1.7 |
| 70 | 68.6 ± 1.8 |

### 4.4. Impurity-Induced Order–Disorder Phase Transition

Figure 6a,b display XRD patterns of Sr-doped calcite samples and Ba-doped calcite samples respectively. With increasing Sr concentration, 113 reflection gradually broadened and finally disappeared at Sr/(Sr + Ca) = 35 mol% as shown in Figure 6a. The same phenomenon was also observed in XRD patterns of Ba-doped calcite samples (see Figure 6b); 113 reflection gradually broadened and disappeared at Ba/(Ba + Ca) = 25 mol%.

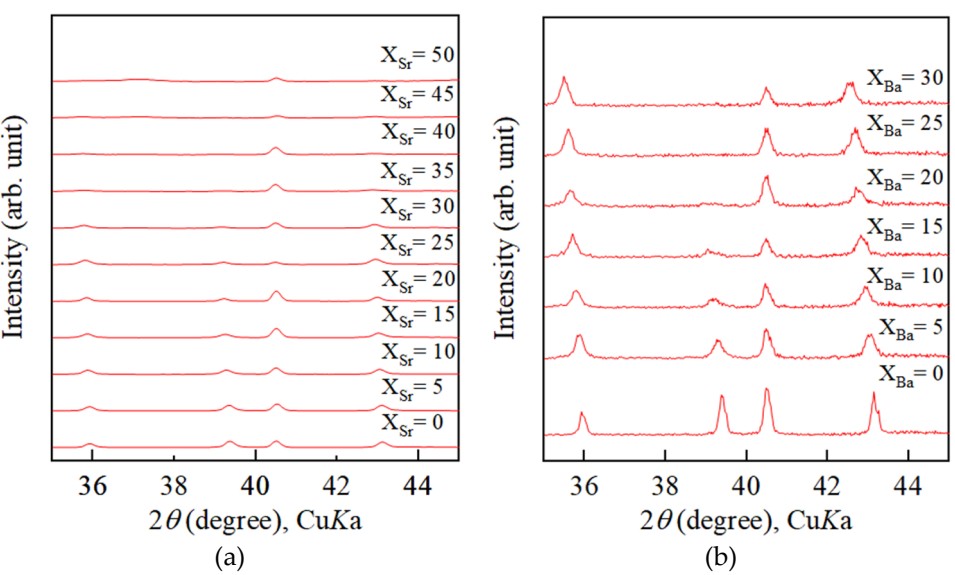

**Figure 6.** Disappearance of 113 reflection observed from powder X-ray patterns of (**a**) Sr-doped calcite samples and (**b**) Ba-doped calcite samples.

At room temperature, $CO_3^{2-}$ in calcite vibrate at the original positions and the alternate layers of $CO_3^{2-}$ are pointing in opposite directions. This is the ordered structure of calcite. In contrast, at high temperature, the $CO_3^{2-}$ in calcite become disordered and leads to the order–disorder phase transition of calcite. This phase transition has been investigated in many studies [18–20]. Dove and Powell made neutron diffraction measurements on high temperature calcite and reported that the intensity of the 113 superlattice reflection falls rapidly on increasing the temperature and then disappears, showing that pure calcite has a phase transition from R$\bar{3}$c to R$\bar{3}$m at 1260 K [18]. Molecular dynamics (MD) simulation of order–disorder phase transition was conducted by Ferrario et al. and Liu et al. [19,20]. They concluded that the equilibrium positions of every $CO_3^{2-}$ occupy the two orientations with equal probability. In addition to these MD simulation studies, Dove et al. studied neutron powder diffraction of calcite and suggested that the order–disorder phase transition is precipitated by the librational amplitude of the $CO_3^{2-}$ [21]. However, the mechanism of this phase transition was still ambiguous. Ishizawa et al. carried out single-crystal X-ray diffraction experiments on high temperature calcite and determined the structure of high temperature calcite called phase V [22]. They reported that 113 reflection completely disappeared at 1275 K, which provides the R$\bar{3}$m symmetry for crystal. In phase V, the oxygen atoms can freely migrate along undulated orbital, which means that carbonate ions are in rotational disorder.

In this study, the extinction of 113 reflection was observed at room temperature in contrast to the previous studies at high temperature. The phenomenon was induced by incorporation of the large incompatible elements which may suggest the rotational disorder of carbonate ions occurring at room temperature. This study is the first to find impurity-induced order–disorder phase transition at room temperature and the detailed mechanism needs to be investigated.

### 4.5. Molecular Dynamics (MD) Simulations of Ba-Doped Calcite

To elucidate the behavior of $CO_3^{2-}$ ions, temperature-dependence of MD-simulated crystals was investigated. Kawano et al. indicated that the relative intensity of the 113–104 reflections decreases gradually upon increasing temperature and abruptly disappears between 1200 and 1250 K. The result is consistent with the experimental results on pure calcite and they suggested that the transition from R$\bar{3}$c to R$\bar{3}$m occurs at this temperature change. Above 1250 K, $CO_3^{2-}$ ions sometimes 'flip' ±60∘ from the original position and changes direction, which corresponds to the rotational (dynamical) disorder of $CO_3^{2-}$ ions. In the same way, behaviors of $Ca_{0.75}Ba_{0.25}CO_3$ and $Ca_{0.5}Ba_{0.5}CO_3$ at high temperature were simulated. The results showed that rotational disorder of $CO_3^{2-}$ ions occurred at 1050 K and 850 K for $Ca_{0.75}Ba_{0.25}CO_3$ and $Ca_{0.5}Ba_{0.5}CO_3$, respectively. The obtained results indicate that the transition temperature notably decreased with increasing Ba concentration in calcite. However, the transition temperatures are significantly higher than room temperature (300 K). This suggests that the rotational disorder associating with free flips of $CO_3^{2-}$ ions does not occur at room temperature even for Ba-containing calcite samples.

Figure 7 shows probability distributions of rotation angle (θ) between the *a*-axis and the C–O bonds for $CaCO_3$, $Ca_{0.75}Ba_{0.25}CO_3$, and $Ca_{0.5}Ba_{0.5}CO_3$ at 300 K. These probability distributions were averaged for the all $CO_3^{2-}$ ions in the simulated cells. All of the O atoms vibrate at the equilibrium positions of R$\bar{3}$c symmetry at 60°, 180°, and 300° from the *a*-axis. With increasing Ba concentration in calcite, the width of the probability distribution increased. However, the probability between the peaks drops to zero. This means that no ±120° flipping of the $CO_3$ groups occur at 300 K.

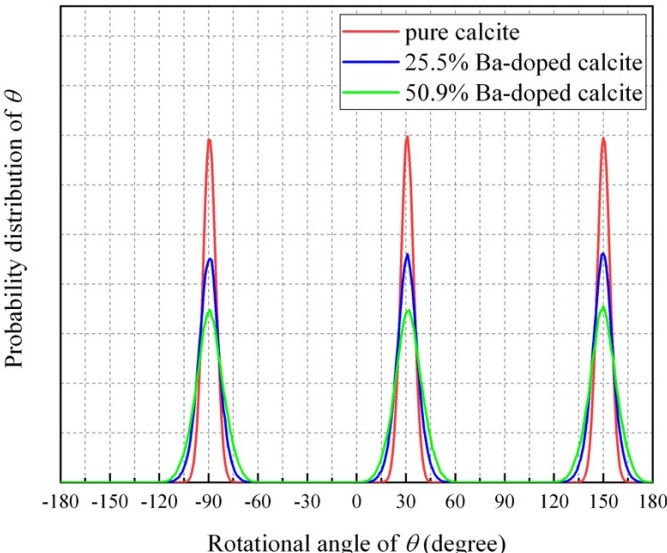

**Figure 7.** Average probability distributions of rotation angle (θ) between the *a*-axis and the C–O bonds for CaCO$_3$, Ca$_{0.5}$Ba$_{0.5}$CO$_3$, and Ca$_{0.5}$Ba$_{0.5}$CO$_3$ at 300 K.

Figure 8 shows probability distributions of rotation angle (θ) between the C–O bonds and the *a*-axis for CaCO$_3$, Ca$_{0.75}$Ba$_{0.25}$CO$_3$, and Ca$_{0.5}$Ba$_{0.5}$CO$_3$ at 300 K. This figure is focusing on one identical CO$_3^{2-}$ ion for CaCO$_3$, Ca$_{0.75}$Ba$_{0.25}$CO$_3$, and Ca$_{0.5}$Ba$_{0.5}$CO$_3$. For pure calcite, three C–O bonds of the CO$_3^{2-}$ ion distribute at −90°, 30°, and 150°. With increasing Ba concentration, and the three C–O bonds angularly displaces from the ideal angles (−90°, 30°, and 150°) and the widths of the distribution increased. Figure 9 shows probability distributions of the libration angle between the *a–b* plane and the CO$_3^{2-}$ ion shown in Figure 8. For pure calcite, the libration angle distributes around 0°. With increasing Ba concentration, the libration angle of the carbonate ion increases and displaces from the equilibrium angle of pure calcite. The simulated results indicate that the involvement of Ba$^{2+}$ ions to calcite induced the displacements of individual CO$_3^{2-}$ ions in the rotational axis (θ) and librational axis (θ). Local structures surrounding a carbonate ion in 25.5% Ba-doped calcite and 50.9% Ba-doped calcite are shown in Figure 9c. The CO$_3^{2-}$ ions in Ba-doped calcite samples vanishing 113 reflection take a static disorder.

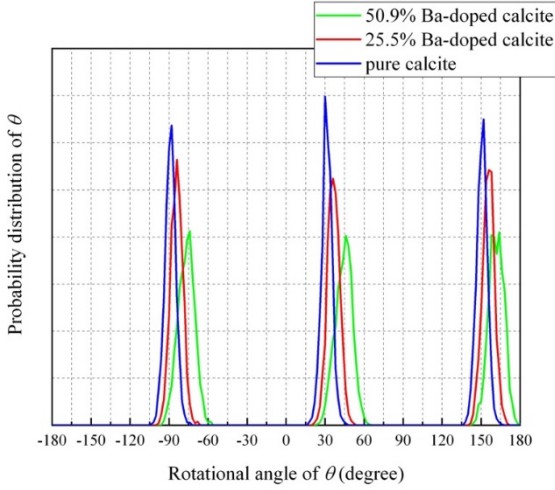

**Figure 8.** Probability distributions of rotation angle (θ) of one CO$_3^{2-}$ ion between the C–O bonds and the *a*-axis for CaCO$_3$, Ca$_{0.5}$Ba$_{0.5}$CO$_3$, and Ca$_{0.5}$Ba$_{0.5}$CO$_3$ at 300 K.

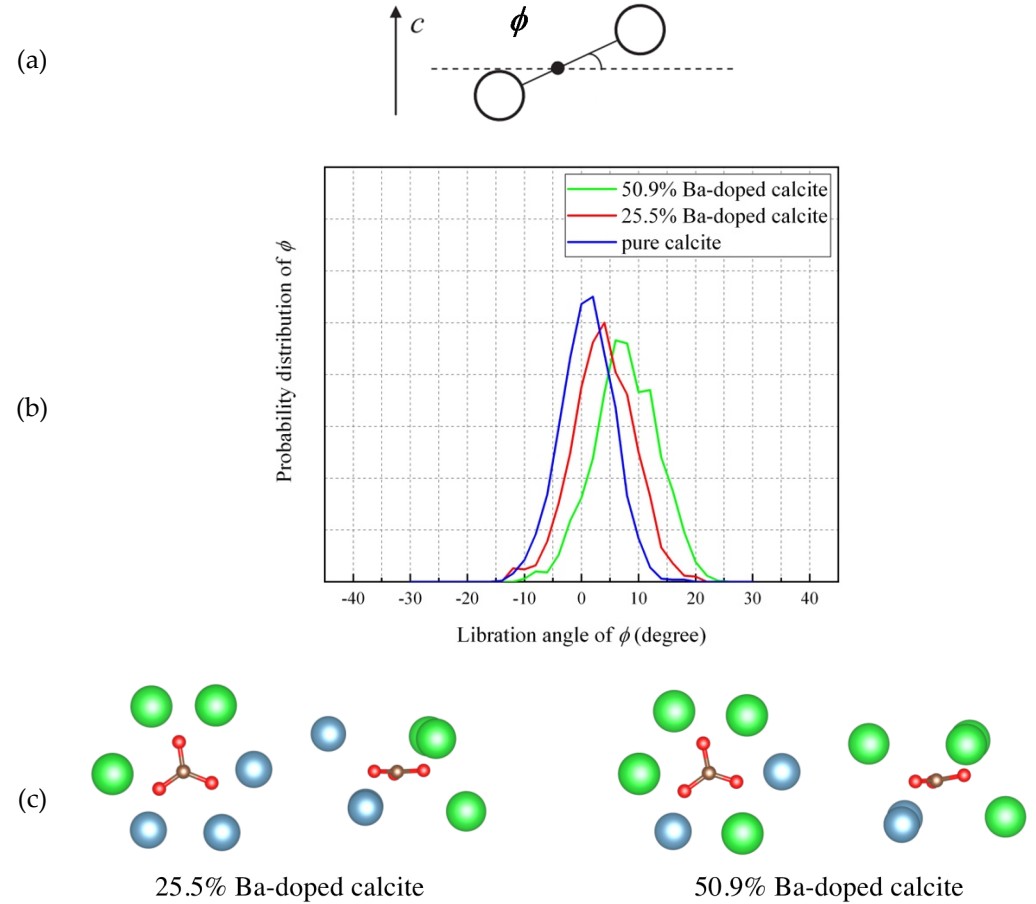

**Figure 9.** (**a**) Schematic drawing of $CO_3^{2-}$ libration by an angle φ out of the *a–b* plane. (**b**) Probability distributions of the libration angle (φ between the *a–b* plane on the same $CO_3^{2-}$ ion shown in Figure 8. (**c**) Local structures surrounding a carbonate ion in 25.5% Ba-doped calcite and 50.9% Ba-doped calcite.

Finally, X-ray diffraction patterns were obtained based on the simulated structure of Ba-doped calcite. As shown in Figure 10, the intensity of 113 reflection decreased with increasing Ba content in calcite and approached zero at Ba/(Ba + Ca) = 0.3. The simulated results are consistent with experimental results obtained in the present work although some difference can be seen between the simulation and the experimental results. The disappearance of 113 reflection in the Ba-doped and Sr-doped calcite samples are derived from the static disorder of $CO_3^{2-}$ ions. This study first clarified the static disorder of $CO_3^{2-}$ ions in calcium carbonate induced by large incompatible cations.

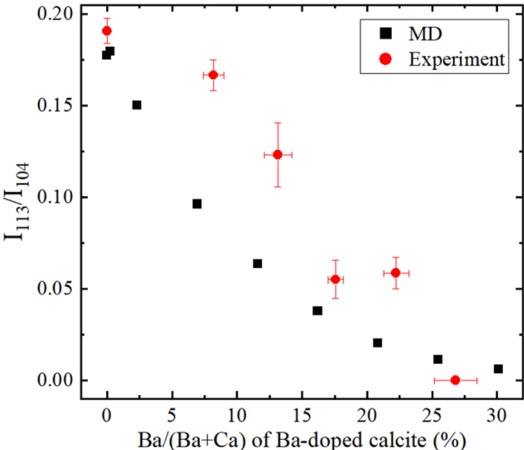

**Figure 10.** Ba concentration in calcite versus reflection intensities of 113 normalized by 104 ($I_{113}/I_{104}$) obtained from the MD simulation and experiments.

## 5. Conclusions

This study clarified that crystallization through an amorphous state can induce the structural incorporation of large and incompatible ions. The introduction of large incompatible ions to calcite induced a newly found phenomenon: the static disorder of carbonate ions. Moreover, the present study may open a window to develop a new material by doping incompatible elements. Specific conclusions obtained from this study are listed below.

(1) Crystallization from Sr-doped amorphous calcium carbonate and Ba-doped amorphous calcium carbonate resulted in the formation of calcite containing notably high concentrations of Sr and Ba. The maximum Ba concentration corresponded to the chemical formula of $Ba_{0.7}Ca_{0.3}CO_3$.

(2) With increasing Sr and Ba concentrations, the intensity of the 113 reflection of calcite decreased. The 113 reflection vanished at room temperature when Ba concentration was higher than 25 mol% ($Ca_{0.75}Ba_{0.25}CO_3$).

(3) The MD simulation indicated that the $CO_3^{2-}$ ions in Ba-doped calcites are not in the rotational (dynamical) disorder but in the static disorder at room temperature. The $CO_3^{2-}$ ions are tilted and angularly displaced from the equilibrium position of pure calcite.

**Supplementary Materials:** The following are available online at http://www.mdpi.com/2075-163X/10/3/270/s1, Figure S1: Ba/(Ba + Ca) molar ratios of starting solutions versus those of ACC samples precipitated from the solutions. Figure S2: (a) Temperature and (b) pressure dependence of the molar volume of MD-simulated and experimentally obtained $BaCO_3$. Black and red arrows in Figure S2(a) shows that phase transitions occurred at that temperature. Figure S3: Cell parameters of MD-simulated and the present experimental results of Ba-doped calcite.

**Author Contributions:** Conceptualization, A.S., H.K., K.K., and J.K.; Experiments, A.S., S.M., D.E., and K.M.; Methodology, A.S., K.K., and J.K.; Supervision, H.K.; Writing—original draft, A.S., H.K., and J.K. All authors have read and agreed to the published version of the manuscript.

**Funding:** The research was supported by JSPS KAKENHI Grant Numbers 18H05224 and JP18K18780.

**Acknowledgments:** The authors would like to thank Hiroshi Sakuma for valuable suggestions on the MD simulation. We are grateful to Takafumi Hirata and Yoshiki Makino for ICP-MS analysis and Hideto Yoshida for technical assistants for SEM-EDS analysis. Comments from three anonymous reviewers greatly helped us to improve this manuscript. We thank Robert C. Liebermann for providing us with an opportunity to contribute a paper to Special Issue of *Minerals* in memory of Orson Anderson.

**Conflicts of Interest:** The authors declare no conflict of interest.

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
