# Peer review of "Incorporation of Incompatible Strontium and Barium Ions into Calcite (CaCO3) through Amorphous Calcium Carbonate"

_minerals, doi:10.3390/min10030270_

Round 1

Reviewer 1 Report

The subject of the manuscript reviewed here is of interest to the readers of Minerals. The manuscript reports on formation of calcite containing high amount of Sr or Ba in process of crystallization from Sr-doped or Ba-doped amorphous calcium carbonates. MD simulations were conducted to investigate the behavior of carbonate ions in Ba-doped calcite. The simulations found that carbonate ions are in static disorder and carbonate ions are tilted and angularly displaced from the equilibrium position of pure calcite.

This is an interesting study and the results are worth to be published in this journal. The paper is clearly organized. However, in order to increase the quality of the study some points should be clarified and developed.

The authors represent their results as quantitative values of Sr and Ba amount in starting solution or Sr and Ba amount in calcite. For expression they used “xSr(or)Ba/ mol ratio %”,  “concentration” or only “Sr/(Ca+Sr)” or “Ba/(Ca+Ba)”. The expression of Sr and Ba amount in starting solution as well as in calcite should be uniform and correct: “xSr/ % in starting solution”, “xBa/ % in starting solution”, “xSr/ % in calcite” and “xBa/ % in calcite”.

xi should be defined as mole fraction x(Ba) = n(Ba)/n(Ba+Ca) or x(Sr) = n(Sr)/n(Sr+Ca) at the beginning of the Results. The uniform expression should be used throughout the whole text, tables, captions, figures and supplements.

In context with this observation, important question is: the method of quantification of Sr or Ba amount in ACC and calcite? Only EDS-SEM? I would use EDS-SEM quantitative but with reservations and supported with other measurements. In manuscript is mentioned also ICP–MS but only for Ba and only for ACC (page 2, raw 78). ICP-MS results should be included in manuscript too and values for Ba as well for Sr in ACC and calcite obtained with this method should be correlated with values from EDS-SEM. Also, the connected question is how did samples prepared for EDS-SEM analysis (homogenization, multiple spectrum…)…more details should be provided in manuscript.

The connected question is regarding sample with x(Sr) = 40 % in starting solution and the result in Table 1a:  this sample is a mixture of calcite and strontianite after ACC heat treatment? (according to manuscript: XRD Figure 1 and text in page 7, raws 126/127: In contrast, ACC samples from solutions with Sr/(Sr + Ca) = 40 mol% crystallized into strontianite and calcite in a similar way to the case of pressure-induced crystallization.). The x(Sr) = 30.7 ± 0.6 % for this sample is Sr content in mixture? Quantification of this mixture (w(calcite)/% and w(strontianite)/%? This results should be included and discuss in the light of observation. Also, with value for Sr content in Table 1a should be note that this sample is a mixture.  

What about x(Sr) = 28.4 ± 0.4 % in sample after ACC pressure treatment? This value is written in text but not in Table 1a. This is a value for samples prepared with pressure treatment of ACC from starting solution with x(Sr) = 40 %? Also mixture of calcite and strontianite? Quantification? Also, needed note in Table 1a near this value and discussion of the results in manuscript.

Other corrections:

- page 2; Experimental procedures; 2.1. Synthesis…; raws 65 – the volume of mixing solutions?

- page 6, raw 162/163 (Figure 4. Caption): instead of “…vs Ba/(Ba+Ca) in ACC determined from ICP-MS.”  should be “…vs Ba/(Ba+Ca) of starting solutions.”

- page 9 (raw 245), page 10 (raws 256, 257; captions Figure 7), page 11 (caption Figure 8): instead of “…for CaCO3, Ca0.5Ba0.5CO3, Ca0.5Ba0.5CO3…” should be “…for CaCO3, Ca0.75Ba0.25CO3 and Ca0.5Ba0.5CO3…”

- page 7, raws 183: instead of “(see Figure 2 and Table 1a)” should be “(see Table1a)”; Figure 2 does not contain information about mol ratio % of Sr in calcite than calcite unite cell volume; the same comment goes to page 7, raw 188 instead of “(see Figure 4 and Table 1b)” should be “(see Table 1b)”.

Author Response

Responses to Reviewer1
The authors represent their results as quantitative values of Sr and Ba amount in starting solution or Sr and Ba amount in calcite. For expression they used “xSr(or)Ba/ mol ratio %”,  “concentration” or only “Sr/(Ca+Sr)” or “Ba/(Ca+Ba)”. The expression of Sr and Ba amount in starting solution as well as in calcite should be uniform and correct: “xSr/ % in starting solution”, “xBa/ % in starting solution”, “xSr/ % in calcite” and “xBa/ % in calcite”.
xi should be defined as mole fraction x(Ba) = n(Ba)/n(Ba+Ca) or x(Sr) = n(Sr)/n(Sr+Ca) at the beginning of the Results. The uniform expression should be used throughout the whole text, tables, captions, figures and supplements.

Thank you very much for the suggestions. Fonts in the figures were unified into Times New Roman.

As suggested, we considered to change expressions for the concentrations of Sr and Ba throughout the manuscript. However, we decided not to change them. xSr or xBa are exclusively used in the figures. This is because the space in the figures are limited. The definitions of the notation were given in the figure captions.

In context with this observation, important question is: the method of quantification of Sr or Ba amount in ACC and calcite? Only EDS-SEM? I would use EDS-SEM quantitative but with reservations and supported with other measurements. In manuscript is mentioned also ICP–MS but only for Ba and only for ACC (page 2, raw 78). ICP-MS results should be included in manuscript too and values for Ba as well for Sr in ACC and calcite obtained with this method should be correlated with values from EDS-SEM. Also, the connected question is how did samples prepared for EDS-SEM analysis (homogenization, multiple spectrum…)…more details should be provided in manuscript.

Thank you very much for asking the questions. We measured Sr and Ba concentrations both in ACC and calcite. However, the mainstream of the discussion in this study is Sr and Ba concentrations in calcite. We moved the descriptions on Ba concentrations in ACC to Supplementary Materials including the analytical procedure. We showed the concentrations of Ba in ACC determined from ICP-MS including the analytical procedure.

We quantified Sr and Ba concentrations in calcite samples using EDS-SEM. Sr and Ba concentrations in ACC samples were determined using ICP-MS. More details for EDS-SEM analysis were added in the main text.

The connected question is regarding sample with x(Sr) = 40 % in starting solution and the result in Table 1a:  this sample is a mixture of calcite and strontianite after ACC heat treatment? (according to manuscript: XRD Figure 1 and text in page 7, raws 126/127: In contrast, ACC samples from solutions with Sr/(Sr + Ca) = 40 mol% crystallized into strontianite and calcite in a similar way to the case of pressure-induced crystallization.). The x(Sr) = 30.7 ± 0.6 % for this sample is Sr content in mixture? Quantification of this mixture (w(calcite)/% and w(strontianite)/%? This results should be included and discuss in the light of observation. Also, with value for Sr content in Table 1a should be note that this sample is a mixture.

We are afraid that our initial manuscript was misleading. The sample prepared by heating ACC precipitated from a starting solution of Sr 40% contained no strontianite. In contrast, the sample mentioned in the main text which contained strontianite was prepared by pressurizing ACC precipitated from a starting solution of Sr 40%. We added an additional sentence “The calcite samples listed in this table contained no strontianite (SrCO3).” in the caption.

What about x(Sr) = 28.4 ± 0.4 % in sample after ACC pressure treatment? This value is written in text but not in Table 1a. This is a value for samples prepared with pressure treatment of ACC from starting solution with x(Sr) = 40 %? Also mixture of calcite and strontianite? Quantification? Also, needed note in Table 1a near this value and discussion of the results in manuscript.

Yes, the Sr concentration (28.4 ± 0.4 % ) given in the main text is for the sample obtained by the pressure treatment on ACC precipitating from a starting solution containing 40 % Sr. We found that the concentration was mistyped and we substituted the correct value (24.8 ± 0.4 %) for the wrong one (28.4 ± 0.4 % ). This value was obtained from SEM-EDS analysis. These samples contained no strontianite. We added these descriptions to the main text.

Other corrections:
- page 2; Experimental procedures; 2.1. Synthesis…; raws 65 – the volume of mixing solutions?
Thank you very much for the suggestion. We mixed the solutions with volume ratio of 1:1. We added the information in the main text.

- page 6, raw 162/163 (Figure 4. Caption): instead of “…vs Ba/(Ba+Ca) in ACC determined from ICP-MS.”  should be “…vs Ba/(Ba+Ca) of starting solutions.”

We corrected the descriptions.

- page 9 (raw 245), page 10 (raws 256, 257; captions Figure 7), page 11 (caption Figure 8): instead of “…for CaCO3, Ca0.5Ba0.5CO3, Ca0.5Ba0.5CO3…” should be “…for CaCO3, Ca0.75Ba0.25CO3 and Ca0.5Ba0.5CO3…”

Thank you very much for pointing out our mistype. We corrected them.

- page 7, raws 183: instead of “(see Figure 2 and Table 1a)” should be “(see Table1a)”; Figure 2 does not contain information about mol ratio % of Sr in calcite than calcite unite cell volume; the same comment goes to page 7, raw 188 instead of “(see Figure 4 and Table 1b)” should be “(see Table 1b)”.
We corrected the descriptions. Thanks.

Reviewer 2 Report

This Reviewer consider that the study performed by authors fulfil the standards of the Minerals journal, but authors need to consider several comments:

In the introduction the authors should highlight the relevance of the study. The reason why the authors carry out this study and its importance in this field is not clear.

In the methodologies section the authors should differentiate the experimental section from the computational one. The authors include the molecular dynamics within the experimental procedures and that is incorrect.

The experimental techniques that have been used in this study (such as ICP-MS, XRD, SEM-EDS, ...) should be explained in the experimental methodology section. All details of each of the techniques (equipment, measurement conditions, etc.) must be specified. Only the X-ray diffraction is explained.

In my opinion, it would be interesting to explain why the synthesis of Sr- and Ba-doped calcite is done in cold and the washing is done with acetone.

On line 78 and 79, they write: “The concentration data indicates that almost all Ba in the starting solutions were accommodated into ACC when precipitated”. I understand that this is a result and therefore should not be in the experimental procedures section.

On the line 91: the two points are double.

In my opinion, the formula on page 3 should have smaller size to be complete on the same line.

In the results the authors could try to improve the quality of figure 5.

In the results, the total time of the molecular dynamics is not given. The duration of the molecular dynamic performed with empirical methods is important. Have the authors ensured that the dynamic has reached equilibrium?

Why did the authors the study of molecular dynamics in Ba-doped calcite and not also in Sr-doped calcite?

In my opinion, the inclusion of a figure with the structures of the models studied with computational methods would be very useful.

The figures included in the results do not have the same type and font size. It is important to unify the type and font size in all the figures of the article.

Finally, it would be important to highlight in the conclusions the relevance of the results obtained.

Author Response

Responses to Reviewer 2
This Reviewer consider that the study performed by authors fulfil the standards of the Minerals journal, but authors need to consider several comments:

In the introduction the authors should highlight the relevance of the study. The reason why the authors carry out this study and its importance in this field is not clear.

Thank you very much for the comment. We have added several sentences to the introduction.

 In the methodologies section the authors should differentiate the experimental section from the computational one. The authors include the molecular dynamics within the experimental procedures and that is incorrect.

We have divided the experimental procedure and molecular dynamics into two independent sections. Moreover, the experimental procedures were subdivided into “Synthesis of Sr-doped calcite and Ba-doped calcite” and “Sample analysis”.

The experimental techniques that have been used in this study (such as ICP-MS, XRD, SEM-EDS, ...) should be explained in the experimental methodology section. All details of each of the techniques (equipment, measurement conditions, etc.) must be specified. Only the X-ray diffraction is explained.

Thank you for the suggestions.

We added descriptions for experimental techniques in more details.

In my opinion, it would be interesting to explain why the synthesis of Sr- and Ba-doped calcite is done in cold and the washing is done with acetone.

We intended to avoid crystallization of calcite by applying cold condition and acetone washing. We added more descriptions to the main text.

 On line 78 and 79, they write: “The concentration data indicates that almost all Ba in the starting solutions were accommodated into ACC when precipitated”. I understand that this is a result and therefore should not be in the experimental procedures section.

We moved these sentences into the Supplementary Materials. More explanations on experimental procedures and techniques have been added.

On the line 91: the two points are double.

One colon (:) was removed.

In my opinion, the formula on page 3 should have smaller size to be complete on the same line.

We changed the font size to smaller one and the equation is now complete in the same line.

In the results the authors could try to improve the quality of figure 5.

Sorry, we have only this quality of data.

In the results, the total time of the molecular dynamics is not given. The duration of the molecular dynamic performed with empirical methods is important. Have the authors ensured that the dynamic has reached equilibrium?

As the reviewer pointed out, the duration of MD simulation is very important. We had ensured that the system had reached the equilibrium during preliminary calculating for 20 ps. Detail explanations for the total time of calculations were added in the main text.

Why did the authors the study of molecular dynamics in Ba-doped calcite and not also in Sr-doped calcite?

Because ionic radius of Ba is notably larger than that of Sr, we can clearly understand the effect of captured divalent cation on the crystal structure by simulating Ba-doped calcite. We explained why we performed Ba-doped calcite in the first paragraph of "Molecular dynamics (MD) simulations of Ba-doped calcite" section.

In my opinion, the inclusion of a figure with the structures of the models studied with computational methods would be very useful.

We added local structures of the model in Figure 9.

The figures included in the results do not have the same type and font size. It is important to unify the type and font size in all the figures of the article.

We unified the font in the figures.

Finally, it would be important to highlight in the conclusions the relevance of the results obtained.

We added several sentences to the conclusions.

Reviewer 3 Report

This manuscript contains both experimental results and MD simulation data, which gives a good story about the Sr and Ba doped calcite structure analysis. However, some modifications are still needed:

Introduction – Please address the importance of introducing Sr2+/Ba2+ into the calcite in the Introduction part. For example, is there any potential application of the Sr2+/Ba2+ doped new material? Experimental procedures – According to the cited literature here, pH plays an important role in ACC preparation. Please indicate the pH value of the mixture in this study. Figure 1(b) – The marker of KCl in figure legend is missing. Please add it. Figure 2 – According to Figure 1(b), there should be a XSr = 0% sample prepared through heating treatment. But the unit cell volume of this sample is not included in Figure 2. Please add it. Also, according to Figure 2, the highest unit cell volume in heating treatment group is higher than 377Å3. But it says 376.51 ± 0.18 in the text. Could authors give some explanations about this? The third question about this plot is the sample size. As the standard deviation is plotted in Figure 2, please indicate where the standard deviation comes from. Is that from different batches of sample or repeated measurements of the same batch of sample? Table 1a – According to Figure 1, there are 10 concentrations (> 0%) of pressure treatment Sr-doped sample and 3 concentrations (> 0%) of heating treatment Sr-doped sample. But only data of 4 pressure treatment samples and 2 heating treatment samples are shown in Table 1a. Is there any reason? Figure 6 – Please move the figures under 3.4 section. 4 Impurity induced order-disorder phase transition – Based on my understanding, the phase transformation discussed here happened under room temperature, correct? In line 225, it says that “…was observed at room temperature”. Could more details be added? Are samples discussed in this section prepared under room temperature? Or, are they pressure/heating treatment sample but stored at room temperature for serval days to let the phase transformation happen? Please give more information.

Author Response

Responses to Reviewer 3
This manuscript contains both experimental results and MD simulation data, which gives a good story about the Sr and Ba doped calcite structure analysis. However, some modifications are still needed:

Introduction – Please address the importance of introducing Sr2+/Ba2+ into the calcite in the Introduction part. For example, is there any potential application of the Sr2+/Ba2+ doped new material?

We have added several sentences in the introduction.

Experimental procedures – According to the cited literature here, pH plays an important role in ACC preparation. Please indicate the pH value of the mixture in this study.

We did not measure the pH value of the mixture in the present study. As pointed out by the reviewer, the pH information is important to conduct experiments reproducibly. The initial pH value of the mixed solution was estimated as 11.5. However, this value drops immediately after mixing solutions because of precipitation of ACC from the solution. We added the pH information in the text.

Figure 1(b) – The marker of KCl in figure legend is missing. Please add it.

Thank you so much for the suggestion. We added the legend in the figure.

Figure 2 – According to Figure 1(b), there should be a XSr = 0% sample prepared through heating treatment. But the unit cell volume of this sample is not included in Figure 2. Please add it.

The plot for XSr = 0% was hidden by the vertical axis. We revised the figure. For the same reason, we revised Figure 4.

Also, according to Figure 2, the highest unit cell volume in heating treatment group is higher than 377Å3. But it says 376.51 ± 0.18 in the text. Could authors give some explanations about this?

The sample with the largest volume higher than 377 Å3 contained strotntianite. The largest cell volume for calcite sample without strontianite was 376.51 ± 0.18 as described in the text. We added comments to the figure caption.

The third question about this plot is the sample size. As the standard deviation is plotted in Figure 2, please indicate where the standard deviation comes from. Is that from different batches of sample or repeated measurements of the same batch of sample?

The error bar was derived from standard deviation of unit cell volume estimated using GSAS. We added the explanation in the figure caption.

Table 1a – According to Figure 1, there are 10 concentrations (> 0%) of pressure treatment Sr-doped sample and 3 concentrations (> 0%) of heating treatment Sr-doped sample. But only data of 4 pressure treatment samples and 2 heating treatment samples are shown in Table 1a. Is there any reason?

Sr concentration was measured on selected samples. There are no special reasons.

Figure 6 – Please move the figures under 3.4 section.

Yes, the figures are under the section.

Impurity induced order-disorder phase transition – Based on my understanding, the phase transformation discussed here happened under room temperature, correct? In line 225, it says that “…was observed at room temperature”. Could more details be added? Are samples discussed in this section prepared under room temperature? Or, are they pressure/heating treatment sample but stored at room temperature for serval days to let the phase transformation happen? Please give more information.

Yes, the extinction of 113 reflection was observed at room temperature. We added in contrast to the previous studies at high temperature” to highlight the results of this study.

Round 2

Reviewer 3 Report

Authors well revised manuscript. I am OK with the updated version.